# Discovery of Novel N-[(dimethylamino)methylene]thiourea (TUFA)-Functionalized Lignin for Efficient Cr(VI) Removal from Wastewater

**DOI:** 10.3390/toxics13090759

**Published:** 2025-09-07

**Authors:** Haixin Wang, Tao Shen, Yiming Wang, Zongxiang Lv, Yu Liu, Juan Wu, Tai Li, Shui Wang, Yanguo Shang

**Affiliations:** 1Jiangsu Provincial Key Laboratory of Environmental Engineering, Jiangsu Provincial Academy of Environmental Science, Nanjing 210036, China; njut_whx@163.com (H.W.);; 2Jiangsu Province Engineering Research Center of Soil and Groundwater Pollution Prevention and Control, Nanjing 210036, China; 3State Key Laboratory of Materials Chemistry, School of Food and Light Industry, Nanjing Tech University, Nanjing 211800, China; 4College of Biotechnology and Pharmaceutical Engineering, Nanjing Tech University, Nanjing 211816, China; shentao@njtech.edu.cn; 5School of Chemical Engineering, Zhengzhou University, No. 100 Science Avenue, Zhengzhou 450001, China; m13770635708@163.com; 6Jiangsu Longhuan Environmental Technology Co., Ltd., Changzhou 210036, China18796929477@139.com (J.W.);

**Keywords:** lignin-based adsorbent, Cr(VI) removal, amine functionalization, wastewater treatment, reusability

## Abstract

This study developed a novel adsorbent for Cr (VI) removal from wastewater by grafting N-[(dimethylamino)methylene]thiourea (TUFA) onto lignin. The resulting TUFA-functionalized lignin adsorbent AL was comprehensively characterized using scanning electron microscopy (SEM), Fourier transform infrared spectroscopy (FT-IR), nuclear magnetic resonance (NMR), and X-ray photoelectron spectroscopy (XPS). Batch adsorption experiments systematically evaluated the influence of solution pH, contact time, temperature, initial Cr (VI) concentration, and adsorbent dosage. AL exhibited high adsorption capacity (593.9 mg g^−1^ at 40 °C), attributed to its abundant nitrogen and sulfur-containing functional groups. Kinetic analysis revealed that the adsorption process followed pseudo-second-order kinetics. Equilibrium isotherm data were best described by the Langmuir model, indicating predominant monolayer chemisorption. Thermodynamic parameters demonstrated that Cr (VI) adsorption onto AL is spontaneous, endothermic, and entropy-driven. The adsorption mechanism involves membrane diffusion and intra-particle diffusion processes. This work successfully synthesized a stable, effective, and low-cost adsorbent (AL) using an amine agent incorporating both nitrogen and sulfur functional groups, offering a promising approach for treating Cr (VI)-contaminated wastewater.

## 1. Introduction

With rapid economic growth, population expansion, and industrial development, heavy metal pollutants in aquatic systems have emerged as a critical global environmental challenge [1,2]. Characterized by persistence, toxicity, and bioaccumulation potential, these pollutants pose severe threats to human health [3]. Chromium, extensively utilized in industries such as electroplating, tanning, pigment manufacturing, and mining, contributes to widespread contamination through uncontrolled discharge of wastewater [4]. In aqueous environments, chromium exists primarily in two valence states: trivalent chromium (Cr (III), an essential trace nutrient that is harmful in excess) and hexavalent chromium [5,6]. Cr (VI) is recognized as a potent carcinogen and mutagen, exhibits high mobility and toxicity, accumulates in biological tissues (e.g., bones), and induces systemic toxicity [7]. The U.S. Environmental Protection Agency (US EPA) has stringent limits for Cr (VI): 0.1 mg·L^−1^ in inland surface waters and 0.05 mg·L^−1^ in drinking water. Consequently, developing effective strategies to reduce Cr (VI) concentrations is necessary.

Conventional methodologies for mitigating hexavalent chromium (Cr (VI)) contamination encompass chemical reduction–precipitation, ion exchange, neutralization, membrane separation, reverse osmosis, flocculation, biological treatment, ultrafiltration, and adsorption [8,9,10]. Each technique, however, exhibits inherent limitations alongside its advantages. For instance, chemical precipitation, while widely implemented, may generate secondary pollution through sludge formation or incomplete contaminant immobilization. In contrast, adsorption is increasingly recognized as a promising approach due to its operational simplicity, lower infrastructure requirements, and the extensive diversity of available adsorbents [11]. Commonly investigated adsorbent materials include activated carbon, functionalized agricultural waste, synthetic polymers, and modified natural polymers. Nevertheless, the development of economically viable and highly efficient adsorbents remains imperative to advance practical, sustainable Cr (VI) remediation.

The production and utilization of biomaterials represents a critical dimension of sustainable development, particularly given their significant potential for mitigating environmental pollution. Lignin, a major structural component of plant cell walls, ranks as the second most abundant biomass resource after cellulose [12]. Despite its ubiquity, lignin remains substantially underutilized as a byproduct of the pulp and paper industries [13]. Over 95% of industrial lignin is discarded as waste or incinerated, contributing to serious environmental contamination. Structurally, lignin contains diverse reactive functional groups including phenolic hydroxyls, aliphatic hydroxyls, methoxy groups, carbonyl groups, and conjugated double bonds [14]. These moieties confer favorable characteristics such as molecular flexibility, tunability, biodegradability, low production costs, and minimal energy loss during processing. Consequently, lignin has garnered increasing attention as an ideal organic polymer component for wastewater treatment applications [15].

To enhance adsorption performance, lignin is frequently modified through chemical functionalization techniques such as sulfonation, etherification, graft copolymerization, crosslinking, and alkylation [16]. The strategic introduction of nitrogen-containing functional groups enhances anion adsorption through complexation and electrostatic interactions, while sulfur-containing groups facilitate cation binding via coordination chemistry, ligand formation, reduction mechanisms, and chelation [17,18,19]. These modifications collectively elevate the contaminant removal capacity of lignin-based adsorbents.

This study synthesized a novel aminated lignin adsorbent (AL) co-functionalized with nitrogen and sulfur heteroatoms, demonstrating robust stability and enhanced adsorption capacity for hexavalent chromium (Cr (VI)). Batch adsorption experiments systematically evaluated key operational parameters affecting Cr (VI) removal efficiency, including solution pH, initial Cr (VI) concentration, contact time, temperature, and adsorbent dosage. Adsorption kinetics and isotherm models were employed to analyze the uptake behavior. Furthermore, reusability assessment confirmed its regenerative potential.

## 2. Material and Methods

### 2.1. Materials

Lignin (Shandong Longli Paper Mill, Dezhou, China), thiourea (Macklin, Shanghai, China), formaldehyde (37 wt% in aqueous solution) (Macklin, Shanghai, China), dimethyl amide dimethyl acetal (DMF-DMA), sodium hydroxide, cyclohexane, and hydrochloric acid were of analytical or biochemical reagent grade and purchased from Aladdin Industrial Co., Ltd, Shanghai, China. All other chemicals used in this study were obtained from commercial sources.

### 2.2. Synthesis of TUFA

Dissolve 1 g of thiourea in 100 mL of ethanol solution and then add 2.1 mL of dimethyl amide dimethyl acetal (DMF-DMA), heated at 75 °C reflux for 3 h; then, spin the mixture and add 30 mL of cyclohexane, stirring for 30 min; the precipitate will be filtered to obtain a light yellow solid powder.

### 2.3. Preparation of TUFA Grafted Lignin

A total of 1 g of lignin dissolved in 100 mL of 0.4 mol L^−1^ NaOH solution was added to a 250 mL eggplant bottle, pH ~13; 0.556 g of TUFA dissolved in deionized water was added to the eggplant bottle, and the mixture was stirred continuously for 30 min to fully dissolve. Add 0.21 mL of formaldehyde solution (38 wt.% aqueous formaldehyde solution) dropwise to the mixture and then increase the temperature to 90 °C for 5 h; after the reaction, cool to room temperature, adjust the pH to neutral with 1 M hydrochloric acid, let it stand, wait for the product to precipitate, centrifuge (8000 r/min, 5 min), lyophilize, and reserve.

### 2.4. Adsorption Studies

The adsorption experiments were performed using a batch method and were used to simulate the adsorption kinetics and adsorption isotherms of Cr (VI) by AL. All intermittent adsorption experiments were carried out in a constant temperature shaker. Various factors, such as pH, adsorbent dosage, temperature, initial Cr (VI) concentration, and contact time, were investigated with respect to Cr(VI) adsorption. Different weights of K_2_Cr_2_O_7_ were dissolved in deionized water to obtain the desired solutions of different concentrations of Cr (VI). The pH of the solution was adjusted with 1.0 M HCl or NaOH solution. The shaker was used to take 0.5 mL of each sample under the same speed conditions. Rotational speed: 180 ppm. The suspension was then centrifuged, and the solution was extracted using a 0.45 mm PTFE syringe filter. To reduce experimental error, three parallel adsorption experiments were performed, and the data given in the text are their average values.

After a period of adsorption, the concentration of Cr (VI) was determined at 542 nm through a UV–Vis spectrophotometer. The adsorption amount (*q*_e_, mg/g) and removal rate (%) were obtained through the following equations:(1)$$q_{e}=\frac{C_{0}-C_{e}}{m}V$$(2)$$q_{e}=\frac{C_{0}-C_{e}}{m}V$$
where *C*_0_ and *C*_e_ (mg/L) are the initial and final concentrations of the Cr (VI) solution, respectively; *V* (L) is the solution volume, and *m* (g) is the biochar weight.

The desorption experiments were performed to study the reusability of the AL. A total of 0.05 g of AL was added to 50 mL of Cr (VI) solution at 25 °C. The recycled AL was dealt with NaOH solution (0.1 M) under 25 °C. After desorption, it was used to adsorb Cr (VI) again.

### 2.5. Characterizations

Nuclear magnetic resonance (NMR) spectra were observed with a Bruker Avance NEO 400 MHz spectrometer (Bruker, Ettlingen, Germany). The elemental analyzer (Thermo, Waltham, MA, USA) was used to determine the content of C, H, N, and S elements. The FT-IR spectra were recorded on a Nicolet iS20 FT-IR spectrometer in the range of 4000 to 400 cm^−1^. Scanning electron microscopy (SEM) images were recorded on a field emission scanning electron microscope (Tescan MIRA LMS, Brno, Czech). X-ray photoelectron spectroscopy (XPS) measurements were conducted on an ESCALAB 250 instrument (Thermo, Waltham, MA, USA) equipped with a Mg Kα X-ray source (hν = 1486.6 eV). UV–Vis absorbance spectra were recorded using a Shimadzu UV-2450PC spectrophotometer (Shimadzu, Kyoto, Japan).

## 3. Results and Discussion

### 3.1. Characteristic Description

#### 3.1.1. Analysis of FTIR

The absorption band observed at 3300 cm^−1^ is attributed to the O-H stretching vibration. Meanwhile, the characteristic peaks located at 2931 cm^−1^, 2840 cm^−1^, and 1460 cm^−1^ are assigned to the C-H stretching vibrations of methyl (-CH_3_), methylene (-CH_2_), and methoxy (O-CH_3_) groups, respectively, while the peaks at 1510 cm^−1^ and 1334 cm^−1^ indicate the guaiacyl and syringyl structural units [20], respectively, and the above indicates that the skeletal structure of lignin was not destroyed during the amination reaction. In addition, the intensity of the peaks at 1510 cm^−1^ and 1334 cm^−1^ in the spectra of AL increased slightly after the amination reaction, which was due to the successful introduction of the amination reagent TUFA containing a methylene group, due to the Mannich reaction occurring at the phenolic hydroxyl active site of lignin, resulting in a reduction of guaiacyl and butyl structural units. The largest differences between the L and AL spectra are at 1124 cm^−1^, 1372 cm^−1^, and 3180 cm^−1^, which are the four C=S, C-N, and N-H vibration-induced absorption peaks (Figure 1). The above analysis confirms that TUFA has been successfully attached to lignin.

#### 3.1.2. Analysis of 2D HSQC

As shown in Figure 2, new signals of δ_C_/δ_H_ (31.1/2.74, 34.44/2.51, and 36.15/2.89) can be observed in the 2D HSQC pattern of AL compared to L. These new signals can be attributed to the methyl and methylene groups in the TUFA structure, and these emerging signal peaks clearly confirm that TUFA has been successfully grafted into lignin. Additionally, the intensity of the signals attributed to G_5_ and H_3,5_ in the aromatic region decreased sharply, indicating that TUFA was mainly grafted to the neighboring or para-regions of the phenolic hydroxyl groups of lignin.

#### 3.1.3. Elemental Analysis

As evidenced by elemental analysis (Table 1), grafting lignin onto TUFA resulted in pronounced increases in elemental content. Specifically, nitrogen content increased from 1.19% to 10.96%, representing a 9.21-fold enhancement, and sulfur content increased from 0.30% to 11.65%, corresponding to a 38.59-fold rise. These results confirm the successful incorporation of TUFA moieties.

#### 3.1.4. Analysis of SEM

Analysis of the SEM images (Figure 3a,b) reveals distinct morphological alterations in lignin following TUFA grafting. The unmodified lignin surface exhibits a relatively smooth texture with limited porosity, a characteristic likely detrimental to effective chromium adsorption. In contrast, the aminated lignin surface displays significantly increased roughness and a distinct lamellar structure. This modified morphology results in a substantially larger comparative surface area, facilitating greater exposure of adsorption sites. Consequently, these structural changes are posited to enhance the chromium adsorption capacity of the aminated lignin. Furthermore, the observed lamellar structure may promote interlayer diffusion of Cr species, while the increased surface roughness could provide additional binding points for complexation or ion exchange processes.

### 3.2. Adsorption of Cr (VI) on AL

#### 3.2.1. Effect of Initial pH

Since the initial pH dictates the protonation state of the adsorbent surface and the ionic form of chromium(VI), evaluating its impact is essential for understanding the adsorption mechanism. Hexavalent chromium exists in aqueous solution mainly in four forms, including Cr_2_O_7_^2−^-, CrO_4_^2−^, HCrO_4_^−^-, and H_2_CrO_4_. The surface charge characteristics of the adsorbent usually gradually changed from positive to negative as the pH of the solution increased. When pH > 6.0, it exists mainly as CrO_4_^2−^, while at pH 2.0–6.0, it exists mainly as HCrO_4_^−^ and Cr_2_O_7_^2−^, and at pH < 1.0, it exists mainly as H_2_CrO_4_. As shown in Figure 4, AL has the most excellent capacity for Cr (VI), with 85.26% removal when pH = 2.0. This is because the reactive groups, such as amine groups, on the surface of AL are easily protonated under acidic conditions, which makes the surface of AL positively charged, and the Cr (VI) anions in solution are adsorbed onto the adsorbent through electrostatic interaction. The protonation of the AL surface decreased with increasing pH. In addition, the predominant form of the presence of Cr (VI) also changed. Thus, the removal of Cr (VI) by AL decreased significantly to 7.63% when the pH was increased from 2.0 to 6.0. In the next adsorption experiments, pH = 2.0 was chosen. At pH values above 7, the surface of AL was negatively charged, and the presence of electrostatic repulsion had a negative effect on the removal rate of Cr(VI), preventing it from being adsorbed.

#### 3.2.2. Effect of Absorption Time

During the first 12 h of adsorption, the Cr (VI) in the solution decreased rapidly, while the removal of Cr (VI) by AL increased (Figure 5). The strongest mass transfer was observed at the early stage of adsorption, and the Cr (VI) in the solution rapidly occupied a large number of active sites on the AL surface. The Cr(VI) removal efficiency is strongly dictated by the contact time. Over the period of 12 to 72 h, the concentration in the solution decreases concomitant with the gradual saturation of binding sites. This saturation diminishes the solid–liquid mass transfer driving force, thereby slowing the adsorption rate. After 72 h, the kinetics are substantially weakened and the system approaches adsorption equilibrium.

#### 3.2.3. Effect of Temperature

To assess the thermodynamic behavior, the adsorption process was studied across four temperatures. Figure 6 demonstrates that elevating the temperature significantly enhanced the Cr(VI) adsorption capacity of AL. For instance, at a 200 mg L^−1^ initial concentration, the capacity increased from 218.7 mg g^−1^ (25 °C) to 321.7 mg g^−1^ (40 °C). This improvement can be explained by the greater thermal energy, which promotes ion mobilization and collision frequency, thereby enabling more chromium ions to access and bind with the adsorbent’s active sites. The direct relationship between temperature and capacity confirms the endothermic characteristic of the adsorption.

#### 3.2.4. Effect of Initial Concentration

It can be seen from Figure 6 that the initial concentration has a great influence on the adsorption capacity of AL. The adsorption capacity of Cr(VI) was 299.3 mg g^−1^ at a low concentration of 150 mg L^−1^. When the initial concentration was increased to 300 mg L^−1^, the adsorption capacity of Cr(VI) reached 367.7 mg g^−1^. The higher the concentration of Cr(VI) in the solution, the higher the adsorption capacity of the adsorbent. The increased adsorption of Cr(VI) on AL at higher concentrations is facilitated by the accompanying rise in potential energy and the stronger mass transfer driving force.

#### 3.2.5. Effect of Adsorbent Dosage

The amount of adsorbent is an important parameter in practical applications, which determines the cost and adsorption energy of the adsorbent. As shown in Figure 7, when the amount of adsorbent was increased from 2 mg to 20 mg, the adsorption amount was much different at the same time, which is due to the high amount of adsorbent and the large number of active sites exposed to the solution, which can interact with Cr (VI). However, the adsorption capacity per unit mass of adsorbent was reduced, probably because a small amount of adsorbent was in the solution, and all active sites on the adsorbent surface were completely exposed, and the aggregation of particles increased with the increase in adsorbent mass, which reduced the effective adsorption surface area, and some adsorption sites were not exposed to chromium ions, resulting in a decrease in adsorption rate. It can also be seen that the adsorption rate increases with the increase in concentration. Therefore, in practical applications, it should be appropriate to increase the amount of adsorbent, which can improve the removal efficiency.

### 3.3. Adsorption Kinetic Study

Pseudo-first-order (PFO) and pseudo-second-order (PSO) kinetic models were used to simulate the experimental data for a better understanding of the mechanism of the adsorption process. Their equations are as follows:


Pseudo-first-order model:

(3)
$$ln\left(Q_{e}-Q_{t}\right)=lnQ_{e}-k_{1}t$$



Pseudo-second-order model:(4)$$t/Q_{t}=\frac{1}{k_{2}Q_{e}^{2}}+\frac{t}{Q_{e}}$$
where *Q_e_* is the amount of Cr(VI) adsorbed at equilibrium (mg g^−1^); *Q_t_* is the amount of Cr(VI) adsorbed at a given time (mg g^−1^); *K*_1_ is the rate constant of first-order adsorption (min^−1^); and *K*_2_ is the rate constant of second-order adsorption (g mg^−1^ min^−1^).

The linear regression coefficients (R^2^) for the PFO model across the four temperatures varied from 0.826 to 0.991, compared to a range of 0.958 to 0.993 for the PSO model (Table 2). With the exception of 25 °C, the PSO model consistently yielded higher R^2^ values than the PFO model. Furthermore, the theoretical adsorption capacities for Cr(VI) derived from the PSO model showed closer alignment with the experimentally determined values. These results indicate that the adsorption kinetics are best described by the pseudo-second-order model, suggesting a chemisorption process as the rate-limiting step. This chemical adsorption likely involves interactions between Cr(VI) ions and nitrogen-, oxygen-, or sulfur-containing functional groups on the AL surface.

The adsorption mechanism may involve both boundary layer and intra-particle diffusion. To identify the dominant mode, Cr(VI) adsorption data obtained at different concentrations were fitted to an intra-particle diffusion model. The equation is as follows:

Weber and Morris kinetic model:(5)$$t/Q_{t}=\frac{1}{k_{2}Q_{e}^{2}}+\frac{t}{Q_{e}}$$
in which *K*_2_ (mg g^−1^ min^−0.5^) means the internal diffusion rate constant of Cr(VI); *C* means the internal diffusion model constant.

As indicated by the linear fitting results of the intra-particle diffusion model (Table 3), the fact that the fitted line does not pass through the origin (C ≠ 0) suggests that intra-particle diffusion is not the sole rate-limiting step. Additional mechanisms may also contribute to controlling the overall adsorption kinetics.

To further understand the adsorption process, the reaction activation energy (*E_a_*) of the adsorption process was calculated using the Arrhenius equation, by which the activation energy can explain whether the adsorption process is dominated by physical/chemical adsorption or kinetic/diffusion-dominated control.

Arrhenius equation:(6)$$K_{2}=Aexp\left(\frac{{-E}_{a}}{RT}\right)$$

*A* is the Arrhenius constant, *E_a_* (J mol^−1^) is the activation energy, R (8.314 J (mol K)^−1^) is the gas constant, and *T* (K) is the absolute temperature.

In general, the activation energy is low for physical adsorption (typically not exceeding 4.20 kJ mol^−1^), while chemisorption has a high activation energy (typically in the range of 8.40 to 83.70 kJ mol^−1^). In addition, kinetically controlled adsorption processes typically have activation energies greater than 25–30 kJ mol^−1^, while those controlled by diffusion have smaller activation energies. The value of activation energy (*E_a_*) of AL is calculated to be 20.1 KJ mol^−1^. This result indicates that the adsorption of Cr(VI) by AL mainly involves chemisorption, and the adsorption process is controlled by diffusion. In addition, the positive value of *E_a_* indicates that the increase in temperature favors the adsorption, and the adsorption process may be a heat absorption reaction.

### 3.4. Adsorption Isotherms Study

The experimental data for Cr(VI) adsorption were fitted to the Langmuir and Freundlich isothermal adsorption models. The mathematical forms of these models are as follows:

Langmuir model:(7)$$\frac{C_{e}}{Q_{e}}=\frac{C_{e}}{Q_{m}}+\frac{1}{bQ_{m}}$$

Freundlich model:(8)$$lnQ_{e}=lnK_{f}+\frac{1}{n}logC_{e}$$
where *Q_e_* is the amount of Cr adsorbed at equilibrium (mg g^−1^); *C_e_* is the equilibrium concentration (mg g^−1^); *Q_m_* is the maximum adsorption capacity (mg g^−1^); *b* is the Langmuir adsorption equilibrium constant. Based on the Langmuir constant, *b*, a dimensionless constant, *K_L_* (the separation factor; L mg^−1^), can be obtained. It can be expressed as K_L_=1/(1+bC_0_); 1/*n* and *K_f_* are the Freundlich equilibrium constants.

Figure 8 shows the fitted graphs of the two models with the individual parameters in Table 4 and Table 5. It can be seen from the graphs that the Langmuir model has a higher R^2^ compared to the Freundlich model, indicating that the adsorption of Cr (VI) on AL is a monolayer adsorption. The basic characteristics of the Langmuir adsorption model are usually described by the separation factor *K_L_*; if *K_L_* > 1 means unfavorable adsorption, *K_L_* = 1 means linear adsorption, 0 < *K_L_* < 1 means favorable adsorption, and *K_L_* = 0 is irreversible adsorption, and the lower the value of *K_L_*, the stronger the affinity between the adsorbent and the adsorbate. At 25, 30, 35, and 40 °C, *K_L_* fulfills the condition of 0 < *K_L_* < 1, indicating that the adsorption process of Cr(VI) on AL is favorable in the studied concentration range. Based on the Langmuir model, the maximum adsorption capacity for Cr(VI) was estimated to be 595.23 mg·g^−1^ at 40 °C. The maximum adsorption capacity of AL (593.9 mg g^−1^) is superior to many previously reported amine-modified adsorbents, highlighting its potential efficacy.

To further understand the adsorption process, the Tempkin model is a chemisorption model based on the interaction between positive and negative charges, assuming that the adsorbent has a non-uniform surface, while dividing the surface into many uniform adsorption units, each with a constant heat of adsorption. The equation is as follows:

Tempkin model:(9)$$Q_{e}=\left(\frac{RT}{b}\right)ln\left(A_{T}C_{e}\right)$$

This equation can be expressed in its linear form as follows:(10)$$Q_{e}=B_{T}lnA_{T}+B_{T}lnC_{e}$$

*A_T_* is the equilibrium binding constant; *B_T_* = (*RT/b*), a plot of *Q_e_* versus *lnCe*, yielded a linear line.

From the data in Table 6 and Figure 9, the adsorption of Cr(VI) by AL is basically in accordance with the Tempkin adsorption model, which indicates that the Cr(VI) adsorbed on the surface of AL is non-uniformly distributed, and the adsorption hot line decreases with the increase in Cr(VI) coverage.

In addition, the adsorption capacity of the AL prepared in this study for Cr(VI) was compared with other lignin-based adsorbents in the recent literature (Table 7). The maximum adsorption capacity of AL for Cr(VI) was found to be better than most of the reported lignin-based adsorbents under the same temperature conditions.

### 3.5. Adsorption Thermodynamics Study

To further investigate the thermodynamic nature of the adsorption process, key parameters—namely, the change in Gibbs free energy (ΔG), enthalpy change (ΔH), and entropy change (ΔS)—were calculated using the following equations:(11)$$K_{d}=\frac{C_{0}-C_{e}}{C_{e}}\times \frac{V}{m}$$(12)$$∆G=-RTlnK_{c}$$(13)$$ln\left(K_{d}\right)=-\frac{∆H}{RT}+\frac{∆S}{R}$$
where *K_d_* (mL g^−1^) is defined as an equilibrium constant, *R* is the gas constant (8.314 J (mol K)^−1^), and *T* is the temperature (K).

The experimental data obtained at a Cr(VI) concentration of 200 mg·L^−1^ were fitted, and the results are summarized in Figure 8 and Table 8. The calculated ΔG values were negative across all temperatures studied, indicating that the adsorption process is spontaneous. In addition, the value of ΔG decreases with increasing temperature, which indicates that increasing temperature is favorable for the reaction. The positive value of ΔS indicates some structural changes in Cr(VI) and AL during the process, and these changes lead to an increase in the confusion at the solid/solution interface. The positive value of ΔH suggests an endothermic adsorption process. The favorable effect of increased temperature on adsorption further supports this observation, consistent with the experimental findings.

### 3.6. Mechanism Study

To elucidate the adsorption mechanism, the adsorbent material was characterized using FTIR, SEM, and XPS both before and after Cr(VI) adsorption. As shown in Figure 1, notable changes occurred in the FTIR spectrum of AL after adsorption (denoted as AL-Cr). Specifically, the peak intensities associated with methyl and methylene stretching vibrations at 2931 cm^−1^ decreased compared to the original AL spectrum. A reduction in peak intensity was also observed at 1460 cm^−1^, corresponding to the C=C stretching in aromatic groups. Additionally, decreases in intensity were detected for the C–N peak at 1372 cm^−1^, the –CH vibration at 2840 cm^−1^, and the C=S peak at 1124 cm^−1^. New peaks emerged at 518 cm^−1^, 804 cm^−1^, and 943 cm^−1^, which are attributed to Cr(VI)–O, Cr(III)–OH, and Cr(III)–O vibrations, respectively. These results indicate that Cr(VI) was reduced to Cr(III) and subsequently adsorbed onto the surface of AL [28]. The changes in these functional groups suggest that they play a crucial role in the Cr (VI) adsorption process.

The adsorption mechanism was further explored by XPS detection. The appearance of Cr 2p in AL-Cr (VI) can be seen in Figure 10a, indicating that chromium ions have been successfully adsorbed onto AL. Each Cr 2p can be divided into two peaks, as shown in Figure 10b; 576.9 and 586.3 eV are Cr (III) 2p1/2 and Cr (III) 2p3/2, respectively, and 578.9 and 587.2 eV are Cr (VI) 2p1/2 and Cr (VI) 2p3/2, respectively, indicating that both Cr (III) and Cr (VI) are on the AL surface. Cr (VI) is 35.21% of the AL surface, and Cr (III) is 64.79% of the AL surface; this indicates that most of the Cr (VI) is reduced to Cr (III), and the low-toxicity Cr (III) coexists with Cr (VI) on the surface of AL. The initial solution contained only Cr (VI), and as the adsorption time increased, the Cr (VI) concentration gradually decreased to equilibrium, and most of it was reduced to Cr (III), and a large amount of Cr (III) was adsorbed on the adsorbent, leaving a small amount of Cr (III) in the solution. The above kinetic, thermodynamic, and infrared results can also be used for demonstrations.

Figure 10c shows the high-resolution energy spectrum of C 1s before and after adsorption. The C 1s peak was divided into three main components: C-C, C-O, and C=O, with binding energies of 284.5 eV, 285.4 eV, and 288.2 eV, and the contents of 57.55%, 34.17%, and 8.28% before adsorption and 58.16%, 31.83%, and 10.01%; the C-O content decreased, and the C=O content increased after adsorption, which indicates that the C-OH group provided electrons for the reduction of Cr (VI) to Cr (III), leading to its own loss of electrons for oxidation to C=O. The high-resolution energy spectrum of O 1s in Figure 10d, attributed to O-H, has binding energies of 532.6 eV, 533.7 eV, and 531.4 eV for C-O and C=O, respectively, also confirming that the C-OH group provides electrons for the reduction of Cr(VI) to Cr(III), leading to its own loss of electrons for oxidation to C=O. A new characteristic peak appears after adsorption at 529.9 eV Cr_2_O_7_^2−^, demonstrating that Cr (VI) is adsorbed on the AL surface in the form of oxygen-containing anions. The binding energies of N=C, N-H, and N-C in N 1s in Figure 10e are 398.9 eV, 399.5 eV, and 400.5 eV, respectively, and it can be seen that N-H is reduced after the adsorption of Cr (VI), and the protonated amino group (N-H^+^) appears at 401.5 eV, which would indicate that oxygen- and nitrogen-containing groups play an important role in the adsorption of Cr (VI), and the amine and oxygen-containing groups are readily protonated in acidic solutions (pH = 2) and thus adsorb Cr (VI) by electrostatic interaction [29]. Figure 10f shows the high-resolution energy spectrum of S 2p before and after adsorption. Before adsorption, the S 2p peaks were mainly two peaks of C=S and S(−II) with binding energies of 164.5 eV and 161.0 eV and contents of 67.94% and 32.06%, respectively. After adsorption, the S(−II) peak disappeared, and the S(VI)/S(IV) peak with a binding energy of 168.9 eV appeared, and its content was 54.43%, while the content of the C=S peak decreased to 45.57%, which would indicate that S(−II) was oxidized by Cr(VI), leading to its own loss of electrons for oxidation to S(VI)/S(IV).

Thus, Cr (VI) is firstly adsorbed on the surface of AL. Cr (VI) is firstly adsorbed on the surface of AL by electrostatic attraction, ion exchange, and complexation. As the adsorption proceeds, part of Cr (VI) is reduced to Cr (III) by redox reactions of adjacent electron donor groups. Then, part of Cr (III) is adsorbed by AL, and the rest is released into the aqueous solution.

### 3.7. Desorption Study

The economic viability of adsorption processes critically depends on adsorbent reusability. Cyclic adsorption–desorption experiments (five cycles; desorption: 0.1 M NaOH, 2 h; adsorption pH = 2) demonstrated robust regeneration of aminated lignin (AL), maintaining 75.6% Cr (VI) removal efficiency (Figure 11). This indicates exceptional retention of AL adsorption capacity. However, progressive depletion of nitrogen, oxygen, and sulfur functional groups occurred with increasing regeneration cycles, attributable to reductive consumption during Cr (VI) treatment. Consequently, a gradual attenuation of Cr (VI) removal efficiency was observed. Despite this mechanistic limitation, AL’s sustained performance establishes its suitability for sustainable Cr (VI) remediation applications where cost-effective regeneration is paramount.

## 4. Conclusions

Chromium (VI) is a highly toxic and carcinogenic heavy metal pollutant prevalent in industrial wastewater streams from electroplating, tanning, and metal finishing operations. Its efficient removal remains a critical challenge due to its environmental persistence, mobility, and severe health risks. Conventional adsorbents like activated carbon or synthetic ion-exchange resins often suffer from limitations such as moderate adsorption capacities, high production costs, limited pH stability, or secondary pollution risks from chemical regeneration, hindering their sustainable large-scale deployment. This study addresses these gaps by successfully synthesizing a novel thiourea-functionalized lignin adsorbent (AL) through grafting N-[(dimethylamino)methylene]thiourea (TUFA) onto a lignin backbone. Its significance extends beyond the remarkable adsorption capacity (593.9 mg·g^−1^) and lies in a dual innovation strategy: “waste valorization” and “precision functionalization”, establishing a paradigm for designing cost-effective, high-performance, and green adsorption materials.

At the material design level, this work profoundly advances the high-value utilization of lignin—an underutilized byproduct of the pulp and paper industry. By chemically modifying inherently inert lignin into a functionally active material via targeted grafting of nitrogen/sulfur-rich groups (confirmed by SEM, FT-IR, NMR, and XPS), this study not only significantly enhances its intrinsic adsorption potential but also overcomes key limitations of conventional biomass-based adsorbents, such as low active site density and poor selectivity. The AL synthesis strategy demonstrates that rational molecular design enables precise manipulation of the surface chemistry of biomass carriers, providing a replicable technical roadmap for the directional functionalization of other abundant biowastes (e.g., agricultural residues)**.**

Regarding adsorption mechanism elucidation, this research transcends mere performance reporting through systematic kinetic, isotherm, and thermodynamic analyses, revealing the micro-scale essence of Cr (VI) removal by AL. The pseudo-second-order kinetic model and Langmuir isotherm fitting confirm chemisorption-dominated monolayer adsorption behavior. Thermodynamic parameters (ΔG, ΔH, and ΔS) fundamentally explain the process’s spontaneity, endothermic nature, and entropy-driven character. Critically, identifying membrane diffusion and intra-particle diffusion as synergistic rate-limiting steps provides direct theoretical guidance for future material optimization, such as tailoring pore structure to reduce diffusion resistance-shifting the design paradigm from empirical trial-and-error to mechanism-informed engineering.

For practical engineering applications, AL’s outstanding value lies in its balanced triad of “performance-cost-sustainability”. Compared to commercial adsorbents, AL leverages lignin, a low-cost, abundant, and renewable industrial residue, as its matrix. Its exceptional adsorption capacity (593.9 mg·g^−1^) combined with broad pH adaptability significantly reduces the adsorbent dosage required per unit of Cr (VI) removed, lowering operational costs. Furthermore, robust reusability (>90% capacity retention after five cycles) ensures economic viability in continuous-flow systems, minimizing waste generation from spent adsorbents. Despite its exceptional adsorption capacity, a significant limitation of the AL adsorbent is its stringent requirement for highly acidic conditions (optimal at pH 2) to achieve maximum efficiency. To enhance the practical viability of AL, future research must prioritize strategies to improve its performance under neutral pH conditions. This could involve further material modifications, such as incorporating additional functional groups with higher pKa values or developing hybrid composites, to maintain a positive surface charge and effective adsorption capacity at a broader, more industrially relevant pH range. Therefore, the combination of high capacity, good reusability, and use of a low-cost feedstock suggests that AL has strong potential for practical application in wastewater treatment.

## Figures and Tables

**Figure 1 toxics-13-00759-f001:**
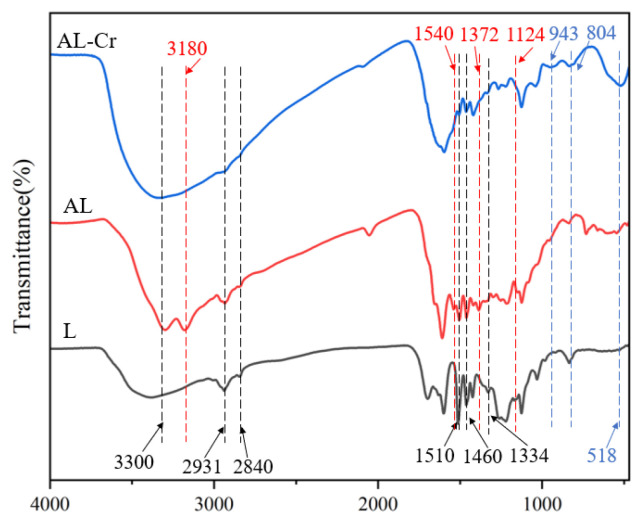
FTIR spectra of raw lignin (L), aminated lignin (AL), and Cr(VI)-laden AL (AL-Cr(VI)) showing characteristic functional groups and changes after adsorption.

**Figure 2 toxics-13-00759-f002:**
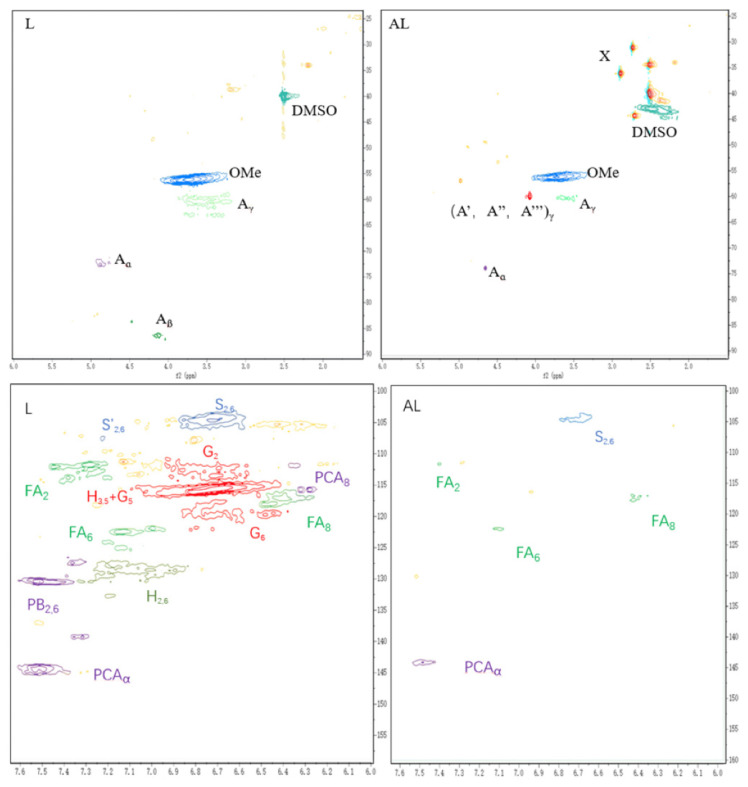
Main classical substructures, involving different side-chain linkages, and aromatic units identified by 2D NMR.

**Figure 3 toxics-13-00759-f003:**
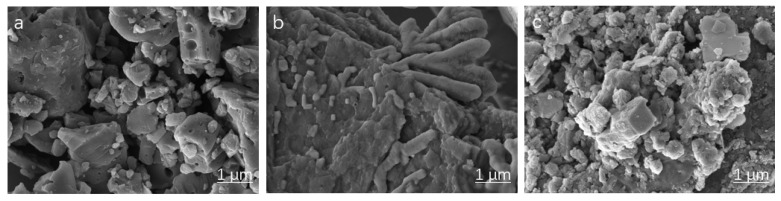
The SEM images of adsorbents: L (**a**), AL (**b**), and AL-Cr(VI) (**c**).

**Figure 4 toxics-13-00759-f004:**
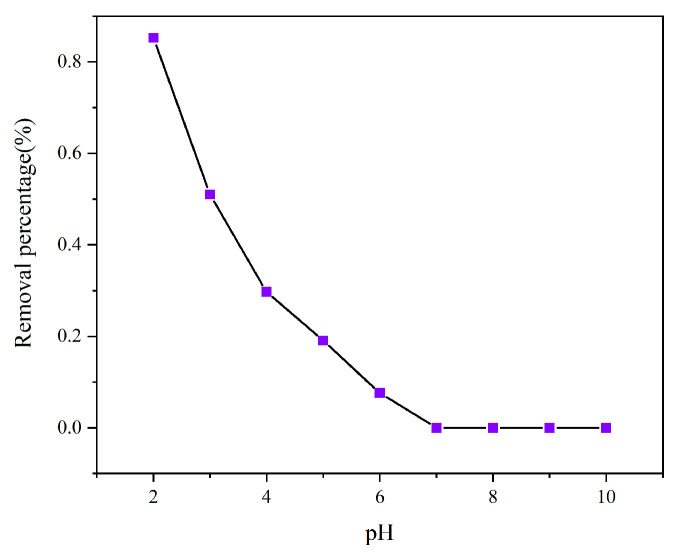
Effect of pH on the adsorption of Cr (VI) on AL.

**Figure 5 toxics-13-00759-f005:**
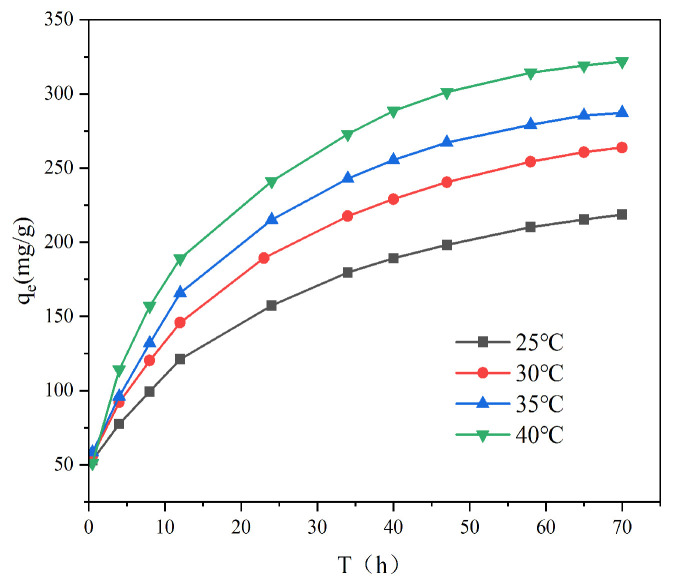
Effect of different temperatures on the adsorption of Cr (VI) on AL.

**Figure 6 toxics-13-00759-f006:**
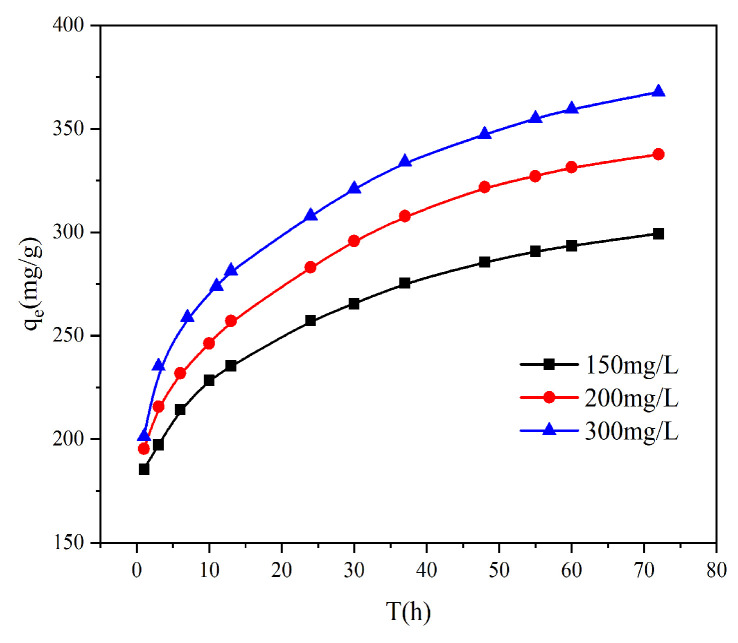
Effect of different initial concentrations on the adsorption of Cr(VI) on AL.

**Figure 7 toxics-13-00759-f007:**
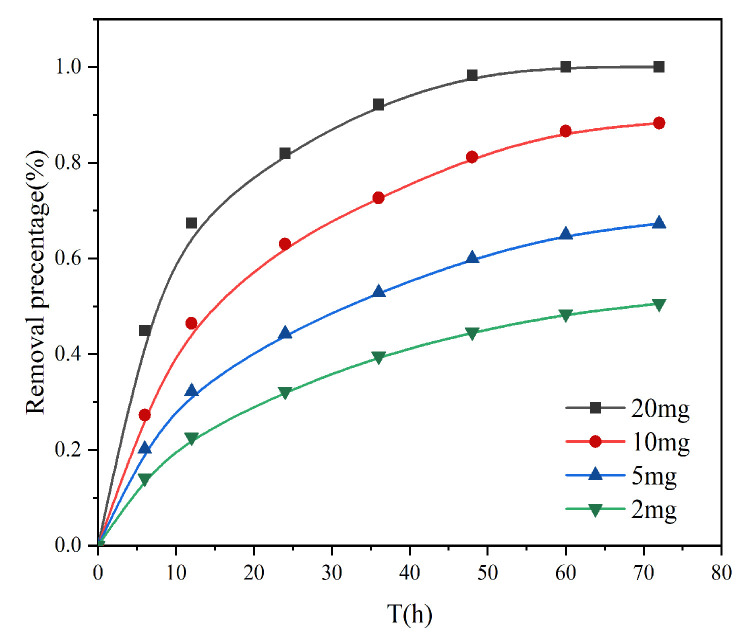
Effect of different adsorbent dosages on the adsorption of Cr (VI) on AL.

**Figure 8 toxics-13-00759-f008:**
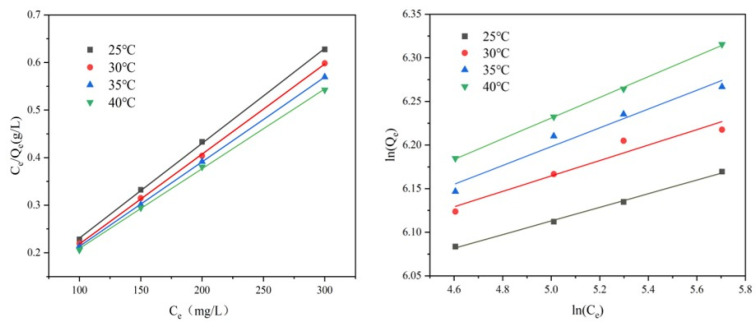
The fitting of Langmuir and Freundlich models.

**Figure 9 toxics-13-00759-f009:**
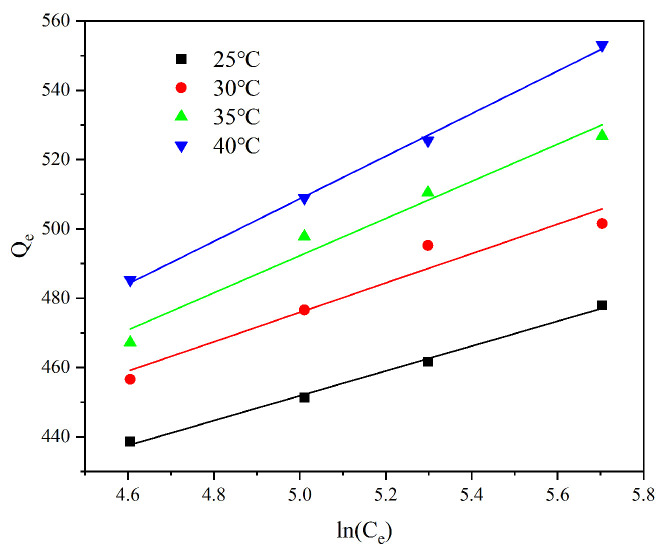
The linear fitting of the Tempkin model.

**Figure 10 toxics-13-00759-f010:**
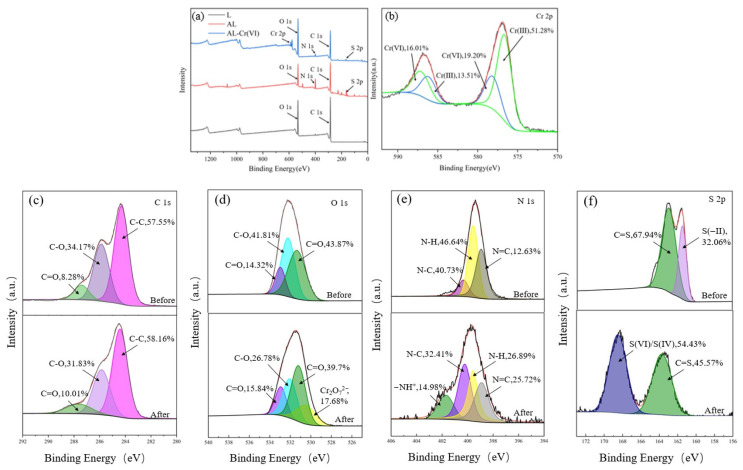
XPS spectra of AL before and after Cr(VI) uptake: (**a**) survey spectrum, (**b**) Cr 2p, (**c**) C 1s, (**d**) O 1s, (**e**) N 1s, and (**f**) S 2p.

**Figure 11 toxics-13-00759-f011:**
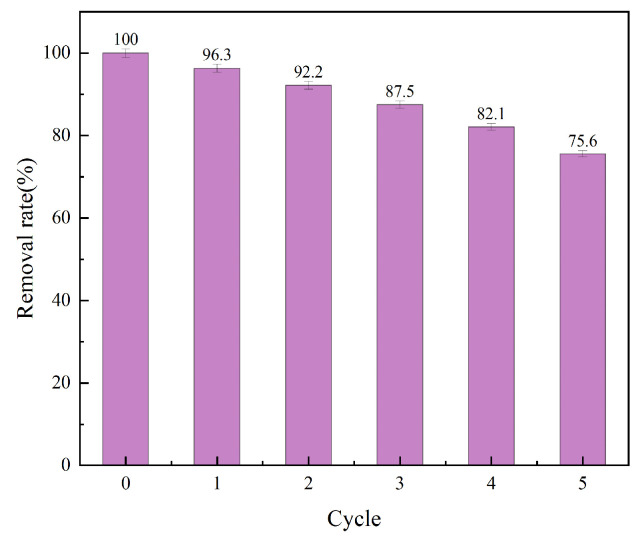
Study on regeneration of AL adsorbent.

**Table 1 toxics-13-00759-t001:** Elemental analysis of L and AL.

Materials	Elements			
	C (wt%)	H (wt%)	N (wt%)	S (wt%)
L	60.15	5.67	1.19	0.30
AL	48.47	5.70	10.96	11.65

**Table 2 toxics-13-00759-t002:** Fitting parameters of kinetic models of Cr(VI) adsorption by AL.

T (°C)	Q_e(exp)_ (mg g^−1^)	Pseudo-First-Order Kinetic Model			Pseudo-Second-Order Kinetic Model		
		K_1_ (min^−1^)	Q_e_ (mg g^−1^)	R^2^	K_2_ (g mg^−1^ min^−1^)	Q_e_ (mg g^−1^)	R^2^
25	501.5	0.00792	514.28	0.991	4.107 × 10^−4^	472.72	0.958
30	528.8	0.00652	431.69	0.932	1.036 × 10^−3^	538.55	0.993
35	561.1	0.0087	413.55	0.826	4.267 × 10^−4^	549.85	0.991
40	593.9	0.02056	528.23	0.973	3.764 × 10^−4^	430.03	0.989

**Table 3 toxics-13-00759-t003:** The Weber and Morris kinetic model for Cr(VI) adsorption on AL at different temperatures.

T (°C)	*K_p_* (mg(g min^1/2^)^−1^)	C	R^2^
25	3.11	28.28	0.996
30	2.29	131.73	0.966
35	4.81	21.81	0.996
40	5.60	17.10	0.994

**Table 4 toxics-13-00759-t004:** Langmuir model parameters for adsorption of Cr(VI) on AL.

T/°C	Langmuir				
	Q_exp_ (mg g^−1^)	Q_e_ (mg g^−1^)	b × 10^3^ (L mg^−1^)	K_L_	R^2^
25	501.5	502.51	63.17	0.0501	0.9997
30	528.8	529.10	64.52	0.0491	0.9997
35	561.1	561.80	50.24	0.0622	0.9999
40	593.9	595.23	40.55	0.1760	0.9994

**Table 5 toxics-13-00759-t005:** Freundlich model parameters for adsorption of Cr(VI) on AL.

Temperature /°C	Freundlich		
	K_F_	1/n	R^2^
25	305.3	0.07831	0.997
30	305.5	0.08852	0.941
35	286.7	0.10789	0.965
40	280.9	0.11855	0.999

**Table 6 toxics-13-00759-t006:** Tempkin model parameters for adsorption of Cr(VI) on AL.

Temperature/°C	Tempkin		
	*B_T_* (J mol^−1^)	*A_T_* (L g^−1^)	R^2^
25	35.85	2.03	0.996
30	42.43	1.83	0.942
35	53.61	1.43	0.971
40	61.45	1.18	0.998

**Table 7 toxics-13-00759-t007:** Comparison of maximum Cr(VI) adsorption capacities of AL with other reported lignin-based and amine-modified adsorbents.

Adsorbents	Adsorption Capacity (mg g^−1^)	References
Organosolv lignin nanoscale zero-valent iron	46.2	[21]
PEI-functionalized kraft lignin	54.20	[22]
PEI-functionalized manganese(IV)oxide nanoparticles	73.9	[23]
PVA-PEI-functionalized magnetite particles	88.4	[24]
A humic acid-coated nitrogen-doped magnetic porous carbon	130.5	[25]
nitrogen-doped porous carbon	481.17	[26]
Hexamethylenediamine-functionalized purified lignin	583.77	[20]
N-methylaniline-functionalized lignosulfonate	1264.8	[27]
TUFA-functionalized lignin	593.9	This paper

**Table 8 toxics-13-00759-t008:** Thermodynamic constants for the adsorption of Cr(VI) on AL at various temperatures.

T (°C)	ΔG (kJ mol^−1^)	ΔH (kJ mol^−1^)	ΔS (kJ mol^−1^ K^−1^)	R^2^
25	−15.07	4.35	65.163	0.9918
30	−15.39			
35	−15.71			
40	−16.05			

## Data Availability

The data generated by this research are all presented in the article.

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
