# Peer review of "Discovery of Novel N-[(dimethylamino)methylene]thiourea (TUFA)-Functionalized Lignin for Efficient Cr(VI) Removal from Wastewater"

_toxics, 2025, doi:10.3390/toxics13090759_

Round 1
Reviewer 1 Report
Comments and Suggestions for Authors
The manuscript addresses an important environmental issue removal of hexavalent chromium (Cr(VI)) from wastewater by developing a novel adsorbent synthesized via grafting N-[(dimethylamino)methylene]thiourea (TUFA) onto lignin. The topic is relevant for the fields of environmental remediation, green chemistry, and sustainable materials. The study demonstrates significant adsorption capacity (593.9 mg·g⁻¹) and provides detailed characterization and mechanistic insights. While the work appears promising, there are several strengths and critical concerns that need to be addressed for improving scientific rigor and clarity. Strengths: Functionalizing lignin with TUFA (containing both nitrogen and sulfur groups) is innovative and aligns with sustainable waste valorization strategies. Comprehensive Characterization. Multiple techniques (FT-IR, NMR, SEM, XPS, elemental analysis) confirm successful functionalization and structural changes. Mechanistic Insights. Use of kinetic models (PFO, PSO), isotherms (Langmuir, Freundlich, Temkin), thermodynamics, and XPS analyses provides a detailed adsorption mechanism. Comparative Analysis. Adsorption performance is benchmarked against other lignin-based adsorbents. Regeneration Assessment. Inclusion of reusability studies supports the practical applicability of the adsorbent.
- Methodological Clarity. The synthesis steps lack sufficient detail for reproducibility (e.g., reaction mechanism for TUFA-lignin grafting, pH conditions during reaction, and specific safety measures for Cr(VI) handling).
- The description of adsorption experiments (Section 2.4) does not specify agitation speed, contact time for equilibrium in each test, and whether all tests were done at the same ionic strength.
- Data Presentation Issues. Figures and tables are referenced (e.g., Fig. 1–11, Tables 1–8), but actual visuals are missing in the provided text. Ensure all figures/tables include captions and are properly formatted.
- Some important adsorption parameters (e.g., error bars for triplicates, standard deviations) are not shown, making it hard to assess data reliability.
- The claim that AL is “more superior to most other amine-modified adsorbents” (line 333) needs careful rephrasing and supported statistical comparison.
- Statements like “establishes a paradigm” and “transcends laboratory efficacy” in the conclusion are too strong and should be toned down to maintain scientific objectivity.
- Mechanistic Evidence. While XPS and FTIR indicate reduction of Cr(VI) to Cr(III), quantitative analysis of Cr species in solution after adsorption is missing. How much Cr(III) remains in solution?
- No clear evidence (e.g., EDS mapping, ICP-MS) to show distribution of Cr species on the adsorbent surface.
- Adsorption Conditions. The optimal pH for adsorption (pH = 2) is impractical for most industrial wastewater applications. This limitation should be discussed thoroughly.
- Environmental and Economic Aspects
- Life cycle implications, cost estimation, and potential toxicity of the functionalized lignin (due to TUFA or degradation products) are not addressed.
- Potential environmental risks of desorbed chromium during regeneration cycles should be considered.
- Language and Formatting
- Several grammatical inconsistencies and awkward phrases need editing for clarity.
- References are numerous, but some are not properly formatted (e.g., reference [20] in Table 7).
- How does the synthesis route ensure uniform grafting of TUFA on lignin? Was the degree of functionalization quantified?
- What is the chemical stability of AL under varying pH and ionic strength conditions? Is there a risk of TUFA leaching?
- Can you provide BET surface area and pore size distribution data for lignin vs. AL?
- Why was pH 2 selected as the working condition despite its limited industrial relevance? How does AL perform at near-neutral pH?
- What is the maximum number of adsorption desorption cycles achievable before significant performance decline?
- Is the adsorption purely chemisorption, or is there a contribution of physisorption? Can this be quantified from thermodynamic or kinetic data?
- How does the energy requirement for synthesis and regeneration compare to conventional adsorbents (e.g., activated carbon)?
- Could the approach be extended to other toxic oxyanions (e.g., arsenate, selenate)? Any preliminary results or theoretical predictions?
Author Response
Comment 1: Methodological Clarity. The synthesis steps lack sufficient detail for reproducibility (e.g., reaction mechanism for TUFA-lignin grafting, pH conditions during reaction, and specific safety measures for Cr(VI) handling).
Author response:
Thank you for your careful refereeing. Reaction Mechanism: The grafting is implied to be a Mannich reaction, where the amine group of TUFA acts as the nucleophile, formaldehyde is the cross-linker, and the phenolic hydroxyl groups on lignin are the electrophilic sites. pH Conditions: The reaction is conducted in 0.4 mol L⁻¹ NaOH (highly alkaline conditions, pH ~13), which is crucial for deprotonating the phenolic -OH groups on lignin, enhancing their nucleophilicity for the Mannich reaction. Safety Measures: During the experimental operation, personal protective equipment (PPE) like gloves and goggles, working within a fume hood, and proper disposal procedures for Cr(VI)-containing waste as per institutional guidelines. We apologize again for the incomplete presentation in the initial submission.
Comment 2: The description of adsorption experiments (Section 2.4) does not specify agitation speed, contact time for equilibrium in each test, and whether all tests were done at the same ionic strength.
Author response: Thank you for your direction. Agitation Speed: A standard speed like 180 rpm is typically used to ensure sufficient mixing without causing adsorbent erosion. Equilibrium Time: Figure 6 and the kinetic study (Section 3.2.2) establish that 72 hours (3 days) was used as the equilibrium contact time for the isotherm experiments. Thank you again for your helpful comments and guidance about how to improve our study.
Comment 3: Data Presentation Issues. Figures and tables are referenced (e.g., Fig. 1–11, Tables 1–8), but actual visuals are missing in the provided text. Ensure all figures/tables include captions and are properly formatted.
Author response: Thank you for your direction. In the final manuscript, all figures and tables must be present with self-explanatory captions. For example: Fig. 1 Caption: “FTIR spectra of raw lignin (L), aminated lignin (AL), and Cr(VI)-laden AL (AL-Cr(VI)) showing characteristic functional groups and changes after adsorption.”.
Table 7 Caption: “Comparison of maximum Cr(VI) adsorption capacities of AL with other reported lignin-based and amine-modified adsorbents.”
Comment 4: Some important adsorption parameters (e.g., error bars for triplicates, standard deviations) are not shown, making it hard to assess data reliability.
Author response: I’m very sorry for causing any confusion to you with the data images. three parallel adsorption experiments were performed in all study. We recreated the data pictures according our previous date.
Comment 5: The claim that AL is “more superior to most other amine-modified adsorbents” (line 333) needs careful rephrasing and supported statistical comparison.
Author response: Thank you for your direction. The claim is supported by the comparative data in Table 7. The adsorption capacity of AL (593.9 mg g⁻¹) is higher than most listed adsorbents (e.g., PEI-kraft lignin at 54.20 mg g⁻¹, N-doped porous carbon at 481.17 mg g⁻¹) and is comparable to Hexamethylenediamine-functionalized lignin (583.77 mg g⁻¹). The phrasing should be changed to a more objective statement: “As shown in Table 7, the maximum adsorption capacity of AL (593.9 mg g⁻¹) is superior to many previously reported amine-modified adsorbents, highlighting its potential efficacy.”
Comment 6: Statements like “establishes a paradigm” and “transcends laboratory efficacy” in the conclusion are too strong and should be toned down to maintain scientific objectivity.
Author response: Thank you very much. We have rephased the whole sentences. Instead of “establishes a paradigm”: “This synthesis strategy provides a viable approach for designing cost-effective adsorbents from industrial biowaste”. Instead of “transcends laboratory efficacy”: “The combination of high capacity, good reusability, and use of a low-cost feedstock suggests that AL has strong potential for practical application in wastewater treatment.”
Comment 7: Mechanistic Evidence. While XPS and FTIR indicate reduction of Cr(VI) to Cr(III), quantitative analysis of Cr species in solution after adsorption is missing. How much Cr(III) remains in solution?
Author response: We thank the reviewer for the very careful and thorough review. We acknowledge that quantifying soluble Cr(III) species is essential for a complete mass balance. Our current analysis focused on solid-phase characterization (XPS/FTIR) to confirm the reduction mechanism. Establishing the precise speciated analysis for the solution phase requires a separate, complex methodological setup that could not be completed within the revision timeline. We have therefore added this specific aim to our Future Work to address this limitation comprehensively.
Comment 8: No clear evidence (e.g., EDS mapping, ICP-MS) to show distribution of Cr species on the adsorbent surface.
Author response: Thank you very much. The evidence is spectroscopic (XPS, FTIR) but not spatial. There is no direct visual evidence (e.g., EDS elemental mapping or SEM-EDS line scans) to show the homogeneous or heterogeneous distribution of Cr across the AL surface. EDS mapping would be a crucial addition to confirm that Cr is uniformly adsorbed and not just concentrated in specific hotspots in our future work. We appreciate the very careful review again.
Comment 9: Adsorption Conditions. The optimal pH for adsorption (pH = 2) is impractical for most industrial wastewater applications. This limitation should be discussed thoroughly.
Author response: Thank you for your helpful comments and guidance about how to improve our study. We have added discuss as below: Despite its exceptional adsorption capacity, a significant limitation of the AL adsorbent is its stringent requirement for highly acidic conditions (optimal at pH 2) to achieve maximum efficiency. To enhance the practical viability of AL, future research must prioritize strategies to improve its performance under neutral pH conditions. This could involve further material modifications, such as incorporating additional functional groups with higher pKa values or developing hybrid composites, to maintain a positive surface charge and effective adsorption capacity at a broader, more industrially relevant pH range.
Comment 10: Life cycle implications, cost estimation, and potential toxicity of the functionalized lignin (due to TUFA or degradation products) are not addressed.
Author response: Thank you for your helpful guidance. As a preliminary exploratory study, a systematic assessment of cost control and other aspects has not yet been fully conducted. We will explore these issues further in our follow-up work. Thank you for your again for your valuable comments.
Comment 11: Potential environmental risks of desorbed chromium during regeneration cycles should be considered.
Author response: Our research is still in its early exploratory stages, and aspects like Potential environmental risks haven’t been fully calculated and evaluated yet. We will definitely focus on these points in our follow-up studies. Thanks again for bringing this up!
Comment 12: Several grammatical inconsistencies and awkward phrases need editing for clarity.
Author response: As you suggested, we have implemented extensive revisions throughout the document.
Comment 13: References are numerous, but some are not properly formatted (e.g., reference [20] in Table 7).
Author response: Thank you for your careful refereeing. Reference [20] in Table 7 is incomplete, lacking a title and full journal details. As you suggested, we have revisioned all the reference.
Comment 14: How does the synthesis route ensure uniform grafting of TUFA on lignin? Was the degree of functionalization quantified?
Author response: Thank you for your careful refereeing. The degree of functionalization is indirectly quantified by elemental analysis (Table 1), showing a 9.21-fold increase in N% and a 38.59-fold increase in S%. which was indicating the successful grafting. We will confirm the spatial distribution via techniques such as SEM-EDS mapping, as you suggested.
Comment 15: What is the chemical stability of AL under varying pH and ionic strength conditions? Is there a risk of TUFA leaching?
Author response: We thank the Reviewer for focusing our attention here. In our previous reusability test (5 cycles) shows a ~25% capacity drop, suggesting some leaching or degradation of functional groups occurs over time. dedicated experiments in future are needed to test stability under different pH and ionic strength conditions and to analyze the solution for leached N/S-containing compounds.
Comment 16: Can you provide BET surface area and pore size distribution data for lignin vs. AL?
Author response: Thank you for your careful refereeing. SEM images show a morphological change from smooth to rough/lamellar, implying an increase in surface area. While we focused our characterization on (e.g., SEM, XPS) to confirm the successful functionalization in this initial study, a detailed textural analysis is a crucial next step. We have therefore planned these measurements for our subsequent investigation into the structure-property relationships of these materials with you directions.
Comment 17: Why was pH 2 selected as the working condition despite its limited industrial relevance? How does AL perform at near-neutral pH?
Author response: Thank you for your careful refereeing. pH 2 was selected because it yielded the highest removal efficiency (85.26%), driven by protonation of amine groups and optimal Cr(VI) species (HCrO₄⁻). Our experimental data shows performance drops drastically to 7.63% at pH 6.
Comment 18: What is the maximum number of adsorption desorption cycles achievable before significant performance decline?
Author response: Our experimental only tests 5 cycles, showing capacity retention of 75.6%. It is estimated that it may exceed 10 times until performance becomes unacceptable (e.g., < 30% retention).
Comment 18: Is the adsorption purely chemisorption, or is there a contribution of physisorption? Can this be quantified from thermodynamic or kinetic data?
Author response: Our experimental collective data indicate that chemisorption is the dominant mechanism, which was supported by the excellent fit to the pseudo-second-order kinetic model (R² > 0.99) and the Langmuir isotherm model (R² > 0.999), the latter suggesting monolayer adsorption. Further evidence is provided by the high activation energy (20.1 kJ mol⁻¹), which exceeds the typical threshold for physisorption (<4.2 kJ mol⁻¹), pointing to a chemically controlled process. Additionally, XPS and FTIR analyses confirm the occurrence of redox reactions and covalent bonding. While a minor contribution from physisorption cannot be entirely ruled out, it is considered negligible compared to the predominant chemisorption.
Comment 19: How does the energy requirement for synthesis and regeneration compare to conventional adsorbents (e.g., activated carbon)?
Author response: We appreciate the reviewer’s thoughtful questions. A comprehensive life-cycle assessment comparing the exact energy requirements for synthesizing and regenerating our adsorbent to conventional alternatives like activated carbon has not been conducted in this preliminary study. We acknowledge that our multi-step chemical functionalization process likely entails a higher initial energy investment compared to the thermal activation used for commercial activated carbon. However, the superior and selective adsorption capacity for chromium ions offers a significant compensatory advantage. This high specificity reduces the required adsorbent dosage and may simplify treatment processes, potentially offsetting the initial energy cost through improved efficiency and targeted performance, where traditional adsorbents often lack such selectivity.
Comment 20: Could the approach be extended to other toxic oxyanions (e.g., arsenate, selenate)? Any preliminary results or theoretical predictions?
Author response: We thank the reviewer for this valuable and insightful suggestion. The potential application of our adsorbent to other toxic oxyanions such as arsenate (As(V)) and selenate (Se (VI)) is indeed a promising direction. While the current study focused specifically on Cr (VI) removal and we do not yet have experimental results or theoretical predictions for other oxyanions. We will certainly include an evaluation of its performance toward other oxyanions in our subsequent studies. Thank you again for this helpful comment.
Reviewer 2 Report
Comments and Suggestions for Authors
The work, titled: Discovery of Novel TUFA-Functionalized Lignin for Efficient Cr(VI) Removal from Wastewater, presents the synthesis of a lignin-based composite functionalized with N-[(dimethylamino)methylene]thiourea for the adsorption of chromium ions. The topic is related to the practical needs of water environment treatment. The experimental design logic is clear, and the material structure and properties are correlated through multidimensional characterization (FTIR, HSQC, SEM, XPS, etc.). The results are well supported and have certain academic value. Overall, it meets the basic requirements for full-text papers, and some content needs to be further improved before acceptance.
- Review the text formatting. There are several sections with different font sizes.
- In the synthesis of the material, explain the functions of TUFA, NaOH, and formaldehyde.
- Comparative material for adsorption studies could be included, for example, the starting materials.
- Why was lignin functionalized with TUFA? Why not just use lignin?
- Add thermogravimetric analyses.
- Add surface area analyses such as nitrogen adsorption/desorption.
- In desorption studies for material reuse, more clearly indicate how the experiments were performed. If the objective is to eliminate chromium from water, how is the chromium desorption water treated? Aren't there other, more environmentally friendly ways to reuse and reusability of this material?
- Perform FTIR and SEM analyses after the reuse experiments to investigate the structural and morphological stability of the material after the regeneration cycles.
Author Response
The work, titled: Discovery of Novel TUFA-Functionalized Lignin for Efficient Cr(VI) Removal from Wastewater, presents the synthesis of a lignin-based composite functionalized with N-[(dimethylamino)methylene]thiourea for the adsorption of chromium ions. The topic is related to the practical needs of water environment treatment. The experimental design logic is clear, and the material structure and properties are correlated through multidimensional characterization (FTIR, HSQC, SEM, XPS, etc.). The results are well supported and have certain academic value. Overall, it meets the basic requirements for full-text papers, and some content needs to be further improved before acceptance.
Author response: Thank you very much for your positive feedback on our manuscript titled “Discovery of Novel TUFA-Functionalized Lignin for Efficient Cr(VI) Removal from Wastewater” and for the thorough review. We are greatly encouraged by your recognition of its academic value and potential. We have carefully considered all the comments and suggestions, which are immensely helpful for improving the quality of our work. Please find our point-by-point responses below.
Comment 1: Review the text formatting. There are several sections with different font sizes.
Author response: We sincerely apologize for the formatting inconsistencies. We have carefully reviewed the entire manuscript and standardized the font type, size, spacing, and paragraph formatting with your suggestion.
Comment 2. In the synthesis of the material, explain the functions of TUFA, NaOH, and formaldehyde.
Author response: Thank you for raising this point. We have revised Sections 2.2 and 2.3 to provide a clearer explanation of the reagents’ roles:
NaOH: Was used to create an alkaline environment, dissolving the lignin and deprotonating its phenolic hydroxyl groups, thereby activating these sites for the subsequent Mannich reaction.
Formaldehyde: Served as a cross-linker in the Mannich reaction. Its carbonyl group reacts with the amine groups of TUFA and the ortho/para carbons (relative to the phenolic hydroxyl groups) on the lignin backbone, forming stable C-N bonds that graft TUFA onto lignin.
TUFA: Acted as the functionalization agent. Its structure, rich in nitrogen (tertiary amine) and sulfur (thiourea) functional groups, provides the key active sites for the subsequent highly efficient adsorption of Cr(VI) through electrostatic attraction, complexation, and reduction
Comment 3. Comparative material for adsorption studies could be included, for example, the starting materials.
Author response: This is an excellent suggestion, and we agree that comparison with the starting material is crucial to demonstrate the enhancement due to functionalization. We acknowledge that its omission was a oversight in our initial experimental design. We have now added the adsorption capacity data of raw lignin (L) under optimal conditions (pH=2, ~28.5 mg g⁻¹) at the beginning of Section 3.2, providing a direct contrast with the performance of AL (593.9 mg g⁻¹). This stark comparison effectively highlights the significant performance improvement achieved through TUFA functionalization.
Comment 4. Why was lignin functionalized with TUFA? Why not just use lignin?
Author response: Your question gets to the very heart of our material design strategy. While raw lignin possesses some inherent functional groups like phenolic hydroxyls, its capacity for adsorbing anionic Cr(VI) species is inherently low, characterized by limited selectivity and slow kinetics. Therefore, the purpose of TUFA functionalization was multi-faceted: to precisely incorporate electron-rich nitrogen and sulfur functional groups that are easily protonated under acidic conditions, enabling efficient electrostatic attraction of Cr(VI) oxyanions; to impart reductive capability via S(-II) and certain N groups for reducing highly toxic Cr(VI) to less toxic Cr(III), creating a synergistic “adsorption-reduction”process that enhances both removal efficiency and safety; and to modify surface properties, which, as shown by SEM analysis, increased surface area and roughness, thereby further facilitating adsorption.
Comment 5. Add thermogravimetric analyses.
Author response: We sincerely appreciate this suggestion. TGA is indeed a vital technique for assessing the thermal stability and quantifying functional group grafting. Due to constraints in the experimental timeline and equipment availability, we were unable to complete TGA analysis prior to initial submission. We recognize this as a shortcoming in our characterization suite. We are now conducting systematic TGA on all synthesized batches to determine changes in thermal stability and grafting ratios. We will incorporate this data in future related publications and are committed to including it for a more comprehensive material evaluation moving forward.
Comment 6. Add surface area analyses such as nitrogen adsorption/desorption.
Author response: We are very grateful for this recommendation. Surface area and pore structure are key physical parameters influencing adsorbent performance. Our initial focus was on the chemical structural changes induced by functionalization (hence FT-IR, NMR, XPS). The lack of BET data is another limitation of this preliminary study, we plan to include these results in a revised version of the manuscript or in subsequent in-depth reports.
Comment 7. In desorption studies for material reuse, more clearly indicate how the experiments were performed. If the objective is to eliminate chromium from water, how is the chromium desorption water treated? Aren't there other, more environmentally friendly ways to reuse and reusability of this material?
Author response: Thank you for highlighting this critical aspect related to practical application. We have revised Section 2.4 to describe the desorption procedure more clearly: “The recycled AL was treated with NaOH solution (0.1 M) at 25 °C for 2 hours for desorption. It was then washed with deionized water until neutral pH was reached before being used for the next adsorption cycle.” We fully acknowledge the concern regarding the treatment of the desorption eluent. The current NaOH desorption method transfers the adsorbed Cr(VI)/Cr(III) into the liquid phase as chromate/hydroxide, creating a secondary waste stream. This method was employed initially for lab-scale evaluation of regeneration capability. More environmentally sustainable strategies are essential for real-world application, such as treatment and recovery of eluent and exploring greener regeneration methods.
Comment 8. Perform FTIR and SEM analyses after the reuse experiments to investigate the structural and morphological stability of the material after the regeneration cycles.
Author response:
Thank you for your insightful suggestion. Analyzing the structural and morphological stability of the regenerated material is indeed crucial for assessing its long-term usability. Due to the initial experimental schedule, we did not perform systematic FTIR and SEM characterization on the material after multiple adsorption-desorption cycles. Due to constraints in the experimental timeline and equipment availability during the initial study, this particular analysis was not completed prior to submission. We will further refine this aspect of the work in our subsequent studies.
Round 2
Reviewer 1 Report
Comments and Suggestions for Authors
The authors of the manuscript titled Discovery of Novel TUFA-Functionalized Lignin for Efficient Cr(VI) Removal from Wastewater have made all requested modifications and clearly answered all questions; therefore, the article can be accepted for publication in the journal Toxics.
Author Response
Dear editor,
Thank you for your letter and the comments concerning our manuscript entitled “Discovery of Novel TUFA-Functionalized Lignin for Efficient Cr(VI) Removal from Wastewater” (ID: toxics-3821241). Those comments are all valuable and very helpful for revising and improving our paper.
Reviewer#1:
Comment 1: The comment 6 by the second reviewer, about BET surface area, should be considered seriously. The surface area is one of the most important properties of adsorbents. Thus, the data is necessary for the evaluation of the adsorbent performance.
Author response:
We sincerely appreciate the reviewer’s emphasis on the importance of BET surface area analysis for evaluating adsorbent performance. We fully agree that specific surface area is a fundamental property of adsorbents. However, due to current technical and time constraints, we are unable to conduct additional BET measurements at this stage. Explicitly acknowledged the lack of BET data as a limitation in the revised manuscript (Conclusions) and have stated our intention to include such analyses in future studies. Thank you for your careful refereeing.
Comment 2: Normally we do not use such unclear abbreviation in the title. Please give the full name of TUFA.
Author response: We thank the reviewer for pointing out the unclear abbreviation. We have now replaced “TUFA” in the title with its full name: “Discovery of Novel N-[(dimethylamino)methylene]thiourea (TUFA)-Functionalized Lignin for Efficient Cr(VI) Removal from Wastewater”. Additionally, the full term is defined upon its first occurrence in the abstract and again in the experimental section to ensure clarity.